# Awareness regarding breast cancer amongst women in Pakistan: A systematic review and meta-analysis

**Muhammad Abdul Rehman** ⬤*, **Erfa Tahir** ⬤, **Huzaifa Ghulam Hussain, Ayesha Khalid, Syed Mohammad Taqi, Eilaf Ahmed Meenai**

Department of Internal Medicine, Dow Medical College, Dow University of Health Sciences, Karachi, Pakistan

* abdurehman528@gmail.com

**Data Availability Statement:** All relevant data are within the manuscript and its Supporting Information files.

## Abstract

### Background

Breast cancer (BCa) is the most common cause of cancer death in Pakistan. In 2019, Pakistan saw the highest global BCa-associated death rate. But do Pakistani women know about the various aspects of BCa? And how prevalent are BCa screening methods amongst Pakistani females? These questions formed the basis for our study.

### Methods

We conducted this review in accordance with Preferred Reporting Items for Systematic Reviews and Meta-Analyses guidelines. On September 1, 2023, we searched PubMed, Embase, Scopus, and Google Scholar, and performed a citation search to search for eligible studies published in 2010 or after, using the following terms: "breast cancer" and "Pakistan". Observational studies that evaluated BCa awareness and/or practice amongst Pakistani females who were not associated with medicine were eligible. We used the National Institutes of Health quality assessment tool to assess the risk of bias. We conducted a proportion meta-analysis to calculate pooled prevalences for variables.

### Results

Responses from 9766 Pakistani women across 18 included studies showed alarmingly low levels of BCa knowledge: risk factors, 42.7% (95% CI: 34.1%-51.4%); symptoms, 41.8% (95% CI: 26.2%-57.5%); diagnostic modalities, 36.3% (95% CI: 23.1%-49.4%); treatments, 46.6% (95% CI: 13.5%-79.8%). Prevalence of breast self-examination (BSE) and ever having undergone a clinical breast exam (CBE) was 28.7% (95% CI: 17.9%-39.6%) and 15.3% (95 CI: 11.2%-19.4%), respectively. BCa knowledge was significantly associated with better educational status, age, and socioeconomic status.

### Conclusion

On average, only two in five Pakistani women are aware of one or more risk factors, symptoms, or diagnostic modalities. Approximately one in two women know about possible BCa

**Funding:** The author(s) received no specific funding for this work.

**Competing interests:** The authors have declared that no competing interests exist.

treatment. Less than one in three women practice regular BSE, and less than one in five women have ever undergone a CBE.

## 1. Introduction

In 2020, more than four million women all over the world lost their lives to cancer [1]. For these deaths, breast cancer (BCa) was the biggest culprit. The mortality associated with BCa is largely concentrated in middle- or low-income countries, which harbor around 72% of cases [1]. This vast difference in BCa mortality between socioeconomic regions has worsened between 1990 and 2019, and the trend is expected to continue [2]. But what do these numbers mean for a low-income, South Asian country like Pakistan?

One in four female cancer-related deaths in Pakistan is attributable to BCa, and the incidence is more than 4.5 times that of the next most common cancer [3]. Between 1990 and 2019, Pakistan has seen a >300% increase in BCa incidence and a 200–300% increase in BCa-associated mortality [2]. As of 2019, along with the Solomon Islands, Pakistan had the highest BCa death rate all over the world [2].

In light of this massive cancer burden, one would expect comprehensive and meticulous systems in place to document and promote BCa awareness in the Pakistani population. Unfortunately, this is not so. Pakistan's cancer-related predicament begins with the National Cancer Registry which is limited in its ability to gauge the actual BCa burden due to obscure statistics, and ends with the lack of national policies to address it [4–6]. According to a recent review, Pakistan has only witnessed one national policy, in its history, to raise awareness regarding cancer's early warning signs [4].

Therefore, with an alarming BCa burden, and shareholding the highest BCa death rate across the world, one would further question how aware are Pakistani women of BCa and its related information. Although several observational studies have sought to answer this question over the years, a consolidated assessment of BCa awareness amongst Pakistani women through a systematic review of the literature was hitherto non-existent.

Synthesis of data through a systematic review of literature is crucial for decision-making and subsequent health policies [7]. In doing so, an assessment of current BCa awareness amongst Pakistani women forms the backbone for future public health-related interventions and/or awareness campaigns. Such health policies are geared towards prevention by teaching the population simple methods to detect and identify BCa through physical examination, or tests like mammography. Inculcation of basic clinical knowledge which is tailored for the common people can help Pakistan catch BCa in its early stages, if not prevent it by addressing modifiable risk factors. Furthermore, by teaching women about available BCa treatments, Pakistan can hope to lose its current spot at the top of the BCa-associated mortality ladder [2].

Therefore, we undertook this study to fill the knowledge gap in the literature regarding BCa awareness amongst Pakistani women. The objectives of our study were to answer the following questions:

1. What is the prevalence of knowledge of BCa in general, its risk factors, symptoms, diagnostic modalities, and treatment?

2. What is the attitude of Pakistani women towards BCa?

3. How prevalent are BCa screening practices amongst Pakistani women?

## 2. Material and methods

We conducted this study in accordance with the Preferred Reporting Items for Systematic Reviews and Meta-Analyses (PRISMA) guideline which is present in **S1 File** [8]. This study was registered at the International Prospective Register of Systematic Reviews (CRD42023452099). Patients or the public were not involved in the design, conduct, reporting, or dissemination plans of this study.

### 2.1 Search strategy

On September 1, 2023, MAR and ET, individually, searched the following databases for relevant studies: MEDLINE, Embase, Scopus, and Google Scholar. The search strings used for each database are present in **S2 File**. MAR and ET also performed a citation search of included studies.

### 2.2 Inclusion/Exclusion criteria

We included observational cross-sectional studies that were published in/after 2010 and satisfied the following criteria: 1) measured BCa awareness, 2) conducted in Pakistan, and 3) the population was women. Studies were excluded based on the following criteria: 1) did not measure BCa awareness, 2) was not conducted in Pakistan, 3) the population was other than women, 4) women who were medical students or physicians, 5) case reports, editorials, letters, and reviews.

### 2.3 Study selection

MAR and ET individually screened titles and abstracts against the inclusion/exclusion criteria to short-list studies for full-text screening. Shortlisted studies underwent full-text review to select studies eligible for inclusion in this study. The PRISMA flowchart is shown in **Fig 1**.

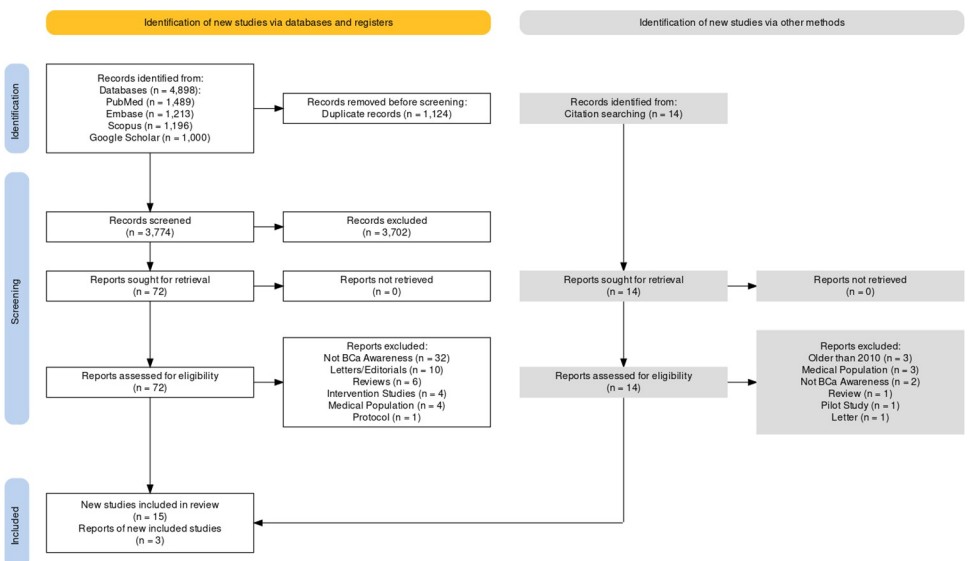

**Fig 1. PRISMA flowchart.** The flowchart shows the databases that were searched and the number of studies that were found, sought, and screened at each step of the screening process.

## 2.4 Data collection

MHG, AK, SMT, and EAM used a predefined form to extract data from all eligible studies in pairs. Extracted results were compared and discrepancies were resolved by a consensus. The following study characteristics were sought: 1) sample size, 2) city/region, 3) mean age, 4) sampling method, 5) questionnaire used. The following data regarding BCa awareness were sought: 1) knowledge about BCa, 2) attitude towards BCa, and 3) practice towards BCa.

## 2.5 Quality assessment

MAR and ET individually applied the National Institute of Health's (NIH) Quality Assessment Tool for Observational Cohort and Cross-Sectional Studies to each study and resolved discrepancies with a consensus [9]. The NIH tool is a 14-item questionnaire that evaluates the integrity of the following characteristics: objective, study population, participation rate, temporal and participant uniformity, sample size, exposure, time frame, effect of varying variables on outcome, validity of variables, frequency of variable assessment, outcome definition, blinding, data loss to follow-up, confounding.

## 2.6 Effect measure

The measure of effect for our study was the pooled prevalence of knowledge about BCa and its associated information: risk factors, symptoms, diagnostic modalities, treatments; and the pooled prevalence of practices associated with BCa.

## 2.7 Statistical analysis

To assess the prevalence of BCa knowledge, we calculated pooled percentages for each variable we evaluated. To account for the varying cities/regions, we used the MetaAnalyst software to assign weights under a random-effects model according to the DerSimonian-Laird method [10]. The $I^2$ characteristic reflected the percentage of variance in the observed effect accountable to the true effect. We assessed publication bias using Begg's rank test and Egger's regression test for funnel plot asymmetry [11, 12]. A $p$-value $<0.05$ was used to signify statistical significance.

## 2.8 Illustrations

We used Canva to build upon the map of Pakistan which was obtained from the Central Intelligence Agency's Map Library. We used MetaAnalyst to generate forest plots for the analysis [10]. The PRISMA flowchart was created using PRISMA2020 [13]. Funnel plots were created using JASP [14].

## 3. Results

We analyzed responses from 9,766 Pakistani women across 18 studies over the past 14 years (2010–2023) [15–32]. The geographical distribution of responses across Pakistan is shown in **Fig 2**.

## 3.1 Characteristics of included studies

All included studies were observational cross-sectional studies. A total of 14 studies used self-made questionnaires, two studies used the Breast Cancer Awareness Measure (BCAM) tool, one used the Breast Cancer Inventory (BCI) tool, and one did not specify [33, 34]. A majority of studies ($n = 10$) had good quality based on the NIH tool. No study showed signs of poor quality although several had concerning copy-writing issues. A detailed quality analysis report is present in **S3 File**.

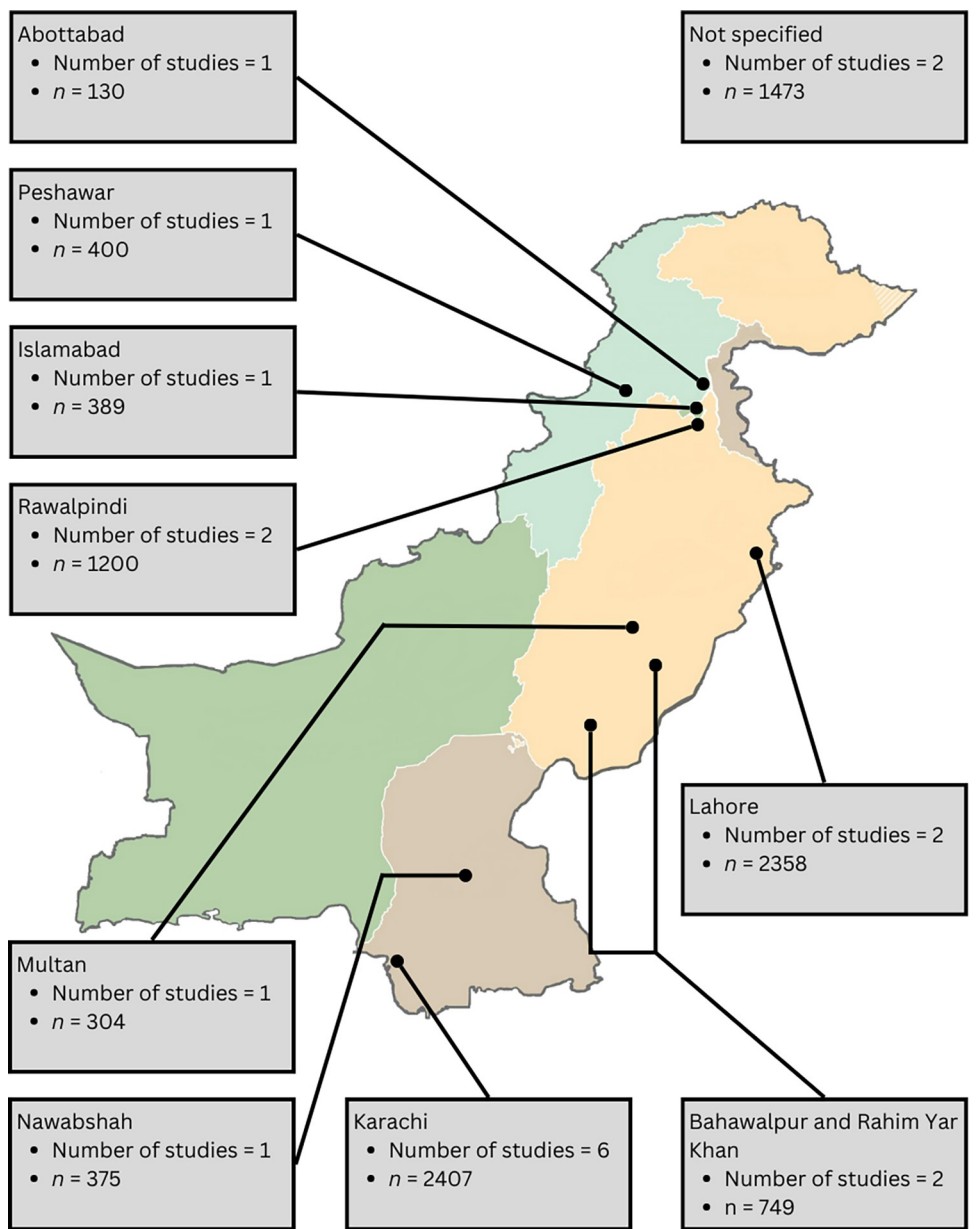

**Fig 2. Geographical distribution of studies.** The map of Pakistan highlights the cities where the included studies were conducted and the number of respondents from each city. This map is similar to the map of Pakistan available at the Central Intelligence Agency's Map Library and is used for illustrative purposes only.

The mean age across studies ranged from 23 years to 35 years. Seven studies evaluated BCa awareness by collecting responses from women who were visiting the outpatient department (OPD), and six studies evaluated responses from non-medical university or college students. Characteristics of included studies are shown in **Table 1**.

## 3.2 BCa knowledge

Out of 18 studies, only seven studies segregated the overall level of BCa knowledge into predefined scores [15, 18, 21, 22, 24, 26, 30]. In four of these studies, the overall BCa knowledge

score was poor for a majority of respondents [15, 21, 24, 26]. A majority had moderate BCa knowledge in the rest [18, 22, 30].

Significant associations of overall BCa knowledge scores were reported between educational status ($n$ = 7) [15, 16, 18, 21, 22, 28, 32], a higher monthly income ($n$ = 6) [15, 18, 21, 22, 28, 32], age ($n$ = 4) [15, 16, 22, 24], and women who were employed ($n$ = 3) [16, 22, 24].

Forest plots for overall knowledge about BCa risk factors, symptoms, diagnostic modalities, and treatment are shown in **Fig 3**. Pooled knowledge for individual variables for each category is highlighted in **Table 2**. Funnel plots for assessment of publication bias are shown in **S4 File**.

### 3.2.A Knowledge of BCa risk factors

The pooled overall BCa risk factor knowledge was 42.7% (95% CI: 34.1%-51.4%). The most commonly assessed risk factors were: family history ($n$ = 13), obesity ($n$ = 12), and nulliparity ($n$ = 10). The least well-known BCa risk factors were: lack of exercise, 24.3% (95% CI: 16.8%-31.7%); menopause at age >50 years, 25.4% (95% CI: 14.8%-36.1%); menarche at age <11 years, 25.7% (95% CI: 16.8%-34.6%); use of hormone-replacement therapy, 26.3% (95% CI, 16.6%-35.9%). The most well-known risk factors were: not breastfeeding, 69.8% (95% CI: 59.2%-80.4%); breast trauma: 67.8% (95% CI: 58.2%-77.5%).

### 3.2.B Knowledge of BCa symptoms

The pooled symptom knowledge was 41.8% (95% CI: 26.2%-57.5%). The most commonly assessed symptoms were: bloody nipple discharge ($n$ = 8), and breast lump ($n$ = 7). The least well-known symptom was the presence of a breast ulcer, 28.8% (95% CI: -2.0–59.5). The most well-known symptom was the presence of a breast lump, 54.7% (95% CI: 32.4–76.9).

Breast pain ($n$ = 7) was considered a BCa symptom by 53.5% (95% CI: 33.1, 73.9) of women.

### 3.2.C Knowledge of BCa diagnostic methods

The pooled knowledge of diagnostic methods was 36.3% (95% CI: 23.1%-49.4%). Mammography was the most frequently ($n$ = 11) evaluated diagnostic method across included studies. The pooled knowledge about mammography as a diagnostic tool was 38.4% (95% CI: 16.3%-60.5%). The least well-known diagnostic modalities were: ultrasound ($n$ = 3), 14.8% (95% CI: 11.6%-17.9%); biopsy ($n$ = 4), 18.6% (95% CI: 4.3%-32.8%).

### 3.2.D Knowledge of BCa treatment

Only four studies evaluated treatment knowledge of BCa [17, 20, 21, 32]. The overall treatment knowledge was 46.6% (95% CI: 13.5%-79.8%). Treatment methods in descending order of popularity were surgery, chemotherapy, and radiotherapy.

### 3.3 Attitude towards BCa

Around 90% of women, across five studies, agreed that they would consult a doctor if they were to discover a breast lump [18–20, 29, 32]. Similarly, the need for BCa screening was also hailed as necessary for early detection of BCa [18, 21, 25, 29, 32].

### 3.3.A Barriers to consultation for breast-related problems

Barriers for women were evaluated in seven studies [15, 16, 19, 21, 23, 29, 32]. The most common barriers to consultation were affordability ($n$ = 5), unavailability of a female doctor ($n$ = 5), and the fear of detecting BCa ($n$ = 5).

**Table 1. Characteristics and quality assessment of included studies.**

| Serial | Author | Study Characteristics | Population Characteristics | n | Quality |
|---|---|---|---|---|---|
| 1 | Shoukat Z *et al.* 2023 | *Study Design*: Cross-sectional<br>*Sampling*: Random Sampling<br>*Questionnaire*: BCAM Tool<br>*Time Period*: September 2019 to February 2021 | *Region*: Many regions/cities<br>*Population*: Females from the general population with no particular criterion.<br>*Mean age ± SD*: 27.4 ± 9.70 years (range, 15–80) | 1000 | Good |
| 2 | Irfan R *et al.* 2021 | *Study Design*: Cross-sectional<br>*Sampling*: Convenience sampling<br>*Questionnaire*: Self-made<br>*Time Period*: January 2019 to April 2019 | *Region*: Nawabshah<br>*Population*: Female pharmaceutical sciences and physiotherapy/rehabilitation university students.<br>*Mean age ± SD*: 21.1 ± 1.2 years (range, 18–26 years) | 375 | Fair |
| 3 | Ullah Z *et al.* 2021 | *Study Design*: Cross-sectional<br>*Sampling*: Systematic random sampling<br>*Questionnaire*: BCAM Tool<br>*Time Period*: November 2017 to April 2018 | *Region*: Peshawar<br>*Population*: OPD patients<br>*Mean age ± SD*: 33.6 ± 12.3 years (range, 18–70) | 400 | Good |
| 4 | Ali A *et al.* 2020 | *Study Design*: Cross-sectional<br>*Sampling*: Consecutive sampling<br>*Questionnaire*: Self-made<br>*Time Period*: May 2018 to July 2018 | *Region*: Karachi<br>*Population*: Women visiting the OPD<br>*Mean age ± SD*: 30.1 ± 7.1 years | 385 | Good |
| 5 | Naqvi AA *et al.* 2018 | *Study Design*: Cross-sectional<br>*Sampling*: Stratified sampling<br>*Questionnaire*: BCI questionnaire<br>*Time Period*: December 2015 to May 2016 | *Region*: Islamabad, Karachi<br>*Population*: Female students and women in general<br>*Mean age ± SD*: NR (range, 18-NR) | 1304 | Good |
| 6 | Arif S *et al.* 2018 | *Study Design*: Cross-sectional<br>*Sampling*: Consecutive sampling<br>*Questionnaire*: NR<br>*Time Period*: December 2016 to June 2017 | *Region*: Karachi<br>*Population*: Female patients' attendants who came to the OPD<br>*Mean age ± SD*: 35.6 ± 9.6 years | 250 | Fair |
| 7 | Rafique S *et al.* 2018 | *Study Design*: Cross-sectional<br>*Sampling*: Convenience sampling<br>*Questionnaire*: Self-made<br>*Time Period*: July 2017 | *Region*: Multan<br>*Population*: Female university students<br>*Mean age ± SD*: 22.6 ± 2.0 years | 304 | Fair |
| 8 | Sultana R *et al.* 2018 | *Study Design*: Cross-sectional<br>*Sampling*: Systematic random sampling<br>*Questionnaire*: Self-made<br>*Time Period*: September 2012 to February 2013 | *Region*: Rawalpindi<br>*Population*: Females visiting the OPD<br>*Mean age ± SD*: 37.5 ± 7.5 years (range, NR) | 200 | Fair |
| 9 | Amin S *et al.* 2017 | *Study Design*: Cross-sectional<br>*Sampling*: NR<br>*Questionnaire*: Self-made<br>*Time Period*: April 2013 to November 2013 | *Region*: Karachi<br>*Population*: Mothers or pregnant females<br>*Mean age ± SD*: NR | 284 | Fair |
| 10 | Masood I *et al.* 2016 | *Study Design*: Cross-sectional<br>*Sampling*: Convenience sampling<br>*Questionnaire*: Self-made<br>*Time Period*: January 2015 to April 2015 | *Region*: Bahawalpur<br>*Population*: Females in general<br>*Mean age ± SD*: 35.2 ± 12.7 years (range, 18–79) | 423 | Good |
| 11 | Noreen M *et al.* 2015 | *Study Design*: Cross-sectional<br>*Sampling*: Systematic random sampling<br>*Questionnaire*: Self-made<br>*Time Period*: NR | *Region*: Bahawalpur and Rahim Yar Khan<br>*Population*: Female medicine and non-medicine university students.<br>*Mean age ± SD*: 23.0 ± 2.1 years (range, 20–28) | 326* | Fair |
| 12 | Sarwar MZ *et al.* 2015 | *Study Design*: Cross-sectional<br>*Sampling*: Convenience sampling<br>*Questionnaire*: Self-made<br>*Time Period*: June 2013 to July 2014 | *Region*: Lahore<br>*Population*: Female patients or their attendants who came to the OPD or were admitted as in-patients<br>*Mean age ± SD*: 32.7 ± 8.7 years (range, NR) | 1184 | Good |
| 13 | Peltzer K *et al.* 2014 | *Study Design*: Cross-sectional<br>*Sampling*: Stratified random sampling<br>*Questionnaire*: Self-made<br>*Time Period*: 2013 | *Region*: Not specified<br>*Population*: Female university students<br>*Mean age ± SD*: NR | 473** | Fair |
| 14 | Raza S *et al.* 2012 | *Study Design*: Cross-sectional<br>*Sampling*: Convenience sampling (for women), snowball sampling (for GPs)<br>*Questionnaire*: Self-made<br>*Time Period*: October 2010 to November 2010 | *Region*: Karachi<br>*Population*: Women and GPs in various towns and colonies<br>*Mean age ± SD*: 36 ± NR years (range, 23–63) | 200*** | Good |

*(Continued)*

**Table 1.** (Continued)

| Serial | Author | Study Characteristics | Population Characteristics | n | Quality |
|--------|--------|----------------------|---------------------------|---|---------|
| 15 | Sobani ZU *et al.* 2012 | *Study Design*: Cross-sectional<br>*Sampling*: Convenience sampling<br>*Questionnaire*: Self-made<br>*Time Period*: November 2010 | *Region*: Karachi<br>*Population*: Female attendants who were accompanying their patients to the hospital<br>*Mean age ± SD*: 32.4 ± 10.9 years (range, NR) | 373 | Good |
| 16 | Khokher S *et al.* 2011 | *Study Design*: Cross-sectional<br>*Sampling*: Convenience sampling<br>*Questionnaire*: Self-made<br>*Time Period*: February 2009 to December 2009 | *Region*: Lahore<br>*Population*: Female non-medicine college and university students<br>*Mean age ± SD*: NR | 1155**** | Good |
| 17 | Ahmad S *et al.* 2011 | *Study Design*: Cross-sectional<br>*Sampling*: Convenience sampling<br>*Questionnaire*: Self-made<br>*Time Period*: July 2010 | *Region*: Abottabad<br>*Population*: Staff nurses working in the hospital<br>*Mean age ± SD*: NR | 130 | Fair |
| 18 | Gilani SI *et al.* 2010 | *Study Design*: Cross-sectional<br>*Sampling*: Stratified random sampling<br>*Questionnaire*: Self-made<br>*Time Period*: January 2009 to May 2009 | *Region*: Rawalpindi<br>*Population*: Female patients or their attendants who came to the OPD or were admitted as patients<br>*Mean age ± SD*: 32.4 ± 10.7 years (range, NR) | 1000 | Good |

*Does not include sample size of medicine students (n = 240)

**Does not include sample size of respondents from other countries (n = 9770)

***Does not include sample size of GPs (n = 100)

****Only 917/1155 were completely filled

**Abbreviations**: BCAM, Breast Cancer Awareness Measurement; BCI, Breast Cancer Inventory; GP, general practitioner; NR, not reported; OPD, out-patient department; RR, response rate; SD, standard deviation.

## 3.4 Practice related to BCa

Practice related to BCa was most frequently reported in the form of breast self-examination (BSE) (*n* = 13), clinical breast exam(s) (CBE) (*n* = 7), and screening by mammography (*n* = 5). The frequency of regular BSE was specified in only four studies, the rest made no clear distinction [15, 16, 19, 31]. Only two of five studies evaluated mammography amongst women aged >40 years, where less than 25% of women older than 40 years had had a mammogram [28, 29].

The pooled prevalence of regular BSE and ever having undergone a CBE were 28.7% (95% CI: 17.9%-39.6%) and 15.3% (95 CI: 11.2%-19.4%), respectively, as shown in **Fig 4**. The distribution of BSE practice across studies is shown in **Fig 5.**

## 4. Discussion

This is the first review of literature that consolidates 14 years of data to examine the level of BCa awareness amongst Pakistani women. Where a plethora of existing literature has highlighted the paucity of BCa awareness amongst Pakistani women at discrete timestamps and locations in Pakistani history, the lack of a consolidated study was a major gap in BCa literature, and our understanding of BCa awareness in South Asia.

Our study shows that the prevalence of knowledge for various aspects of BCa is very low: risk factors, 43%; symptoms, 42%; diagnostic modalities, 36%; treatment, 47%. Only 38% of women knew mammography as a diagnostic tool. The prevalence of regular BSE is much lower with only 29% of Pakistani women performing regular BSE. Furthermore, only 15% of women have ever undergone a CBE.

It is worthwhile to discuss the poor knowledge among Pakistani women about mammography as a diagnostic tool. Over 30 years ago, the benefit of BCa screening by mammography was established [35]. Although some debates have arisen regarding the benefits and harms of

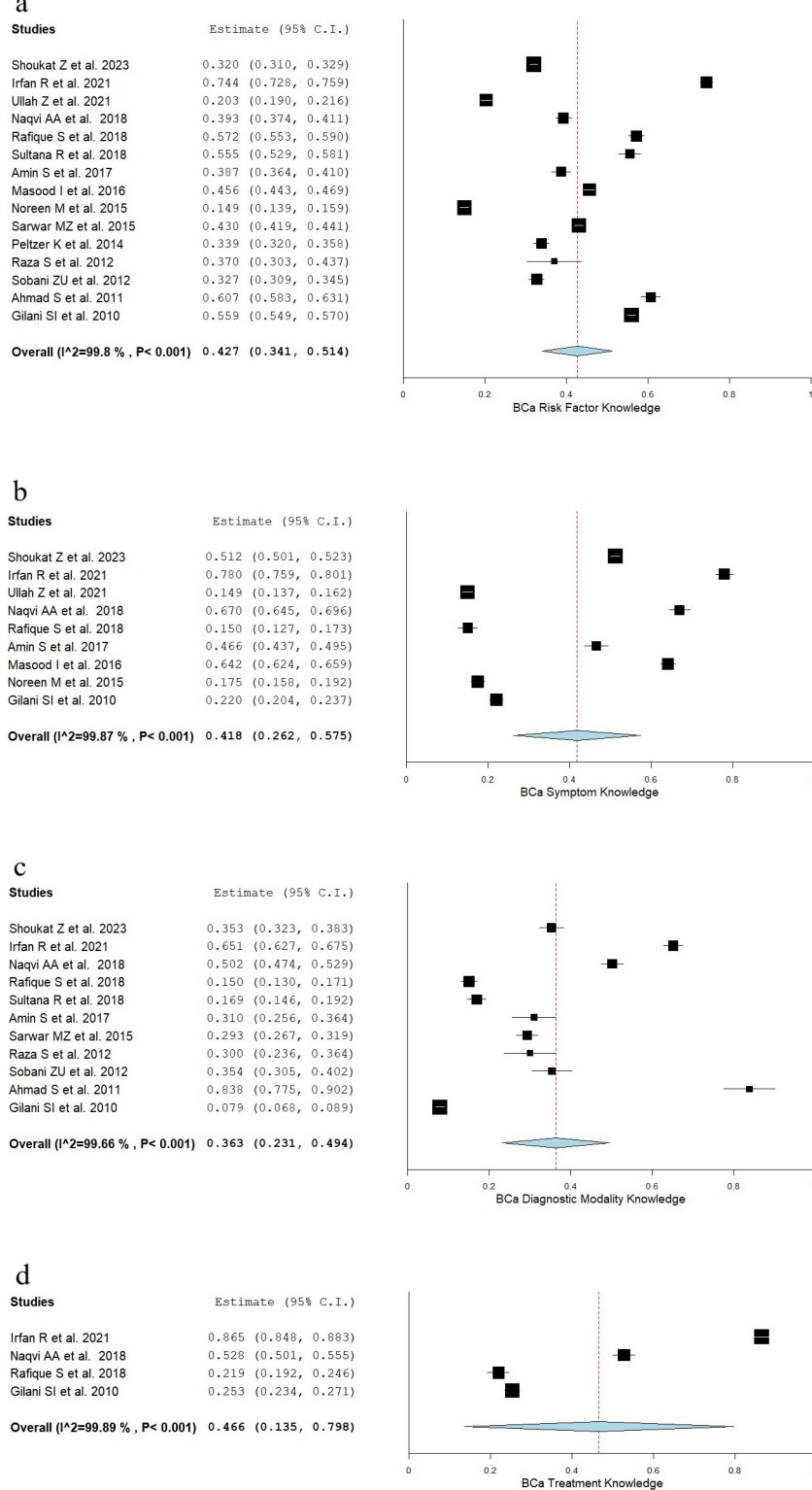

**Fig 3. Forest plots for BCa Knowledge.** These forest plots represent the results of the proportional meta-analysis where data were pooled for the following aspects of BCa: a) risk factors, b) symptoms, c) diagnostic modalities, and d) treatments.

**Table 2. Pooled knowledge of individual BCa risk factors, symptoms, diagnostic modalities, and treatments.**

| | Number of Studies | Average Percentage (range) | Weighted Percentage (95% CI) |
|---|---|---|---|
| **Risk Factor** | | | |
| **Non-Modifiable** | | | |
| Family history of BCa | 13 | 42.9% (25.9–73.6) | 45.4% (36.1, 54.8) |
| Personal history of BCa | 4 | 46.2% (22.4–69.2) | 46.6% (29.9, 63.2) |
| Increasing age | 9 | 52.4% (13.5–88.0) | 54.5% (37.3, 71.6) |
| Breast lump(s) | 3 | 67.1% (21.5–81.2) | 57.7% (23.9, 91.5) |
| Menarche at age <11 years | 9 | 26.2% (6.7–47.7) | 25.7% (16.8, 34.6) |
| Menopause at age >50 years | 6 | 20.6% (5.8–52.0) | 25.4% (14.8, 36.1) |
| **Modifiable** | | | |
| Breastfeeding | 6 | 70.7% (47.0–92.3) | 69.8% (59.2, 80.4) |
| First child-birth at age >30 years | 7 | 35.3% (6.7–60.0) | 31.9% (17.7, 46.2) |
| Nulliparity | 10 | 28.1% (5.8–60.0) | 30.6% (22.1, 39.1) |
| Smoking | 8 | 55.9% (18.4–80.0) | 54.9% (35.5, 74.3) |
| Alcohol | 4 | 40.3% (20.9–67.2) | 40.8% (24.3, 57.4) |
| Obesity | 12 | 42.7% (21.0–80.8) | 43.3% (33.6, 53.1) |
| OCPs | 9 | 50.2% (4.0–73.2) | 47.0% (27.0, 67.0) |
| Trauma | 6 | 66.2% (54.0–78.8) | 67.8% (58.2, 77.5) |
| HRT | 4 | 27.0% (16.3–42.6) | 26.3% (16.6, 35.9) |
| Exercise | 3 | 26.7% (19.0–30.9) | 24.3% (16.8, 31.7) |
| **Symptom** | | | |
| Breast lump | 7 | 56.8% (17.0–93.9) | 54.7% (32.4, 76.9) |
| Breast ulcer | 3 | 22.9% (7.0–65.3) | 28.8% (-2.0, 59.5) |
| Morphological breast change | 5 | 46.7% (14.6–88.0) | 46.4% (20.9, 71.9) |
| Morphological nipple change | 5 | 34.8% (9.5–46.8) | 31.3% (15.7, 46.9) |
| Bloody nipple discharge | 8 | 39.3% (4.5–77.3) | 36.9% (15.3, 58.4) |
| Neck/armpit lump | 5 | 43.8% (11.7–72.8) | 39.8% (16.3, 63.2) |
| Breast pain* | 7 | 55.4% (17.4–90.9) | 53.5% (33.1, 73.9) |
| **Diagnostic Modality** | | | |
| Mammography | 11 | 35.9% (5.3–96.3) | 38.4% (16.3, 60.5) |
| Ultrasound | 3 | 14.5% (13.3–18.4) | 14.8% (11.6, 17.9) |
| Biopsy | 4 | 15.9% (5.1–46.9) | 18.6% (4.3, 32.8) |
| Chest X-ray | 2 | 47.0% (5.0–69.3) | 37.1% (-25.9, 100.2) |
| Blood/urine test(s) | 3 | 35.8% (16.4–48.0) | 35.6% (14.1, 57.0) |
| **Treatment** | | | |
| Surgery | 3 | 59.7% (20.4–91.9) | 51.6% (6.4, 96.8) |
| Chemotherapy | 3 | 42.5% (24.4–85.9) | 47.1% (5.6, 88.5) |
| Radiotherapy | 3 | 28.5% (9.1–76.5) | 33.3% (-5.1, 71.8) |

*Interpret with caution. Breast pain in the absence of other symptoms is not a typical symptom of BCa.

**Abbreviations**: BCa, breast cancer; CI, confidence interval; HRT, hormonal replacement therapy; OCP, oral contraceptive pill.

mammography, the International Agency for Research on Cancer, in 2014, concluded that mammography holds an overall net benefit for women [36]. However, the merits of such a tool may be questionable in a country such as Pakistan given that only two in five women are aware of mammography, as highlighted in our study.

Low levels of physical examination practices are also dismaying. Physical examinations are cheap, convenient, and easy to perform. CBEs by trained medical professionals help detect BCa, although the sensitivity of the exam might be low [37, 38]. The use of BSE is also debated

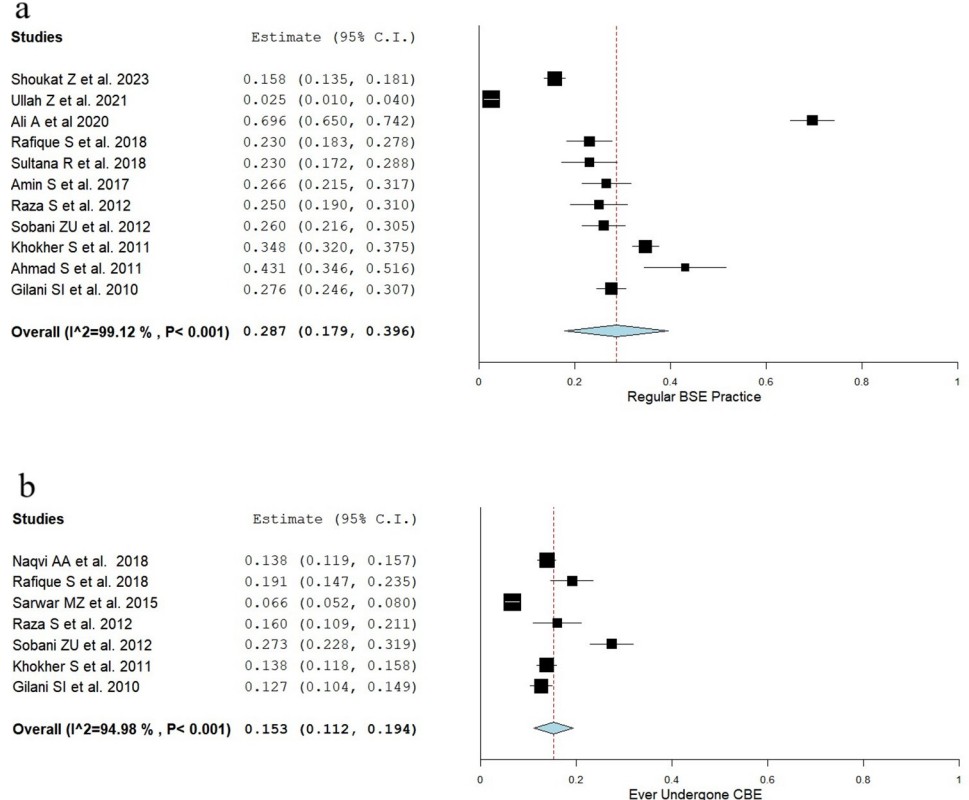

**Fig 4. Forest plots for BCa practice.** These forest plots represent the results of the proportional meta-analysis where data were pooled for the following aspects of BCa: a) regular BSE, b) having ever undergone a CBE.

concerning its false-positive results that encourage unnecessary testing, no established effect on BCa mortality, and the anxiety-provoking psychological stress when a lump is detected—for it may not be cancer at all [39]. However, similar to CBEs, BSEs also help detect BCa, and their use is encouraged in other parts of the medical literature [40]. This holds especially true

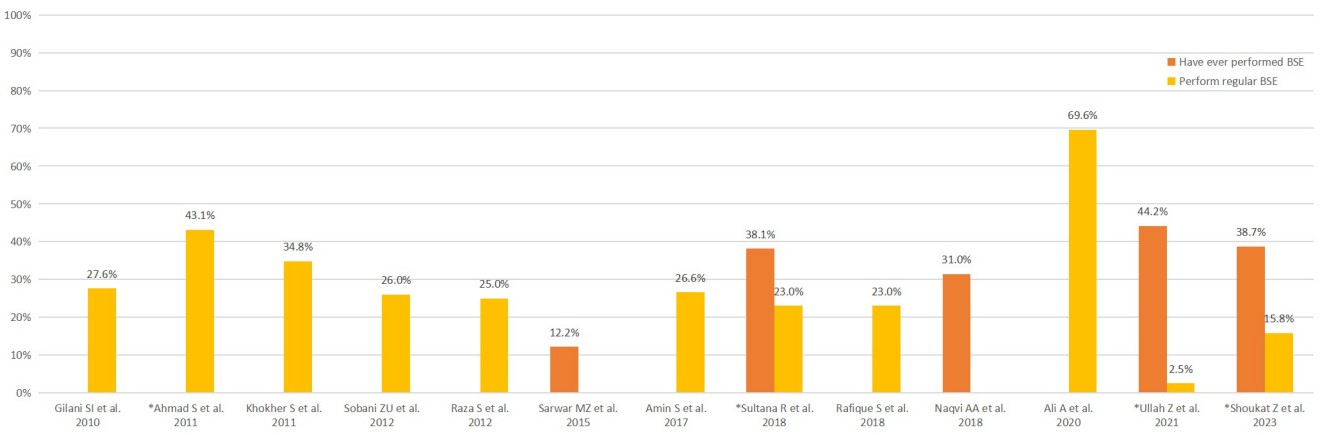

**Fig 5. BSE practice.** This clustered bar chart shows the proportion of women practicing BSE as reported in the included studies. Studies with an asterisk (*) specified the frequency of regular BSE as "monthly".

for low-income countries like Pakistan, where mass screening programs through mammography testing would be hugely expensive. In Pakistan, BSE could be the only option for many women who do not have access to mammography [39, 41].

We would like to address the common misconception of breast pain as a potential symptom of BCa. A recent study of almost 11,000 referrals of women to the breast diagnostic clinic, showed no association of breast pain and BCa [42]. Referrals based on breast pain alone were quoted as "*inefficient*" with "*no health benefit*" but "*increased costs*" [42]. This information is hardly groundbreaking; breast pain, in the absence of other BCa symptoms, is not an indication for referral to the BCa unit [43, 44]. However, our study revealed that breast pain was the second most commonly associated symptom with BCa, with approximately 54% of women harboring this belief. This grim statistic underscores the poor knowledge and awareness women in Pakistan have regarding BCa.

An important trend that should not be missed in the chronological forest plots is the stagnant prevalence of BCa knowledge and practice over the years. The unwavering levels of BCa awareness could largely be attributed to Pakistan's struggling cancer care which, in the face of considerable issues such as a preferential budget that withholds money from healthcare, poor governance, corruption, lack of resources, etc., has largely lacked organized BCa awareness campaigns [6, 45]. The foundation for cancer prevention lies in educating the population at risk about the disease and what they can do to prevent it through screening. Examples of strategically planned BCa awareness campaigns in the medical literature establish their effectiveness [46, 47]. Such interventions inculcate knowledge about risk factors and symptoms that women should look out for. Coupled with awareness about screening practices, BCa awareness interventions reduce late diagnoses, which translates into reduced mortality [48].

The need for interventions that seek to spread BCa awareness, extends beyond the border into India. BCa screening practices are comparable between the two countries as highlighted by the findings of a recent review in India and our study: BSE, 27% vs. 29%; CBE, 17% vs. 15% [49]. Low BCa awareness amongst Asians has also been corroborated by another recent meta-analysis [50]. Another important takeaway from the results of our study and the aforementioned reviews is the significant association between BCa awareness and the level of education or age [49, 50]. Smaller original studies from Bangladesh [51], Afghanistan [52], and Nepal [53], also highlight the dire need for interventions to improve the situation in South Asia, though a consolidated review is absent from the literature.

We observed a large variation in observed results which were attributable to heterogeneity in the true results, highlighted by high $I^2$ values (>90%). In light of the nature of this study, where we pooled prevalence data from multiple observational studies, high $I^2$ values are the norm where they signify a low sampling error which is attributable to the inclusion of several studies, and many individual events [54]. In addition, although our study focused on one country, we assessed data from nine cities and respondents that differed in their education levels, socioeconomic statuses, age, the year the study was carried out, etc., which could explain the high heterogeneity in the true results [55].

We also used funnel plots to detect publication bias in our studies. We observed asymmetrical distributions for knowledge about BCa symptoms, diagnostic modalities, and treatment. A symmetrical distribution around the effect size was observed in funnel plots for knowledge of BCa risk factors, prevalence of BSE and CBE, which was confirmed with the Begg's and Egger's tests which did not show significant funnel plot asymmetry. However, we remind and caution interpreters regarding the subjectivity of the funnel plots [56], the debated use of Begg's and Egger's tests in proportion meta-analyses [55], and the limited power of the tests when few studies are included in the analysis [57]. Furthermore, because asymmetry in funnel plots of proportion meta-analyses can be caused by reasons other than publication bias [56], the

asymmetry in aforementioned analyses could be attributed to the fewer number of studies that evaluated the symptoms ($n = 8$), diagnostic modalities ($n = 11$), and treatments ($n = 4$). We should also not forget the use of self-developed questionnaires to collect responses which could also play a role in biased data collection methods, leading to varying effect sizes in individual studies, and in turn, dispersed plots.

Our study has some limitations. The use of varying questionnaires limited the uniformity of data in the included studies. The use of self-developed questionnaires also introduced bias in reported data which is attributable to how questions are phrased and how responses are interpreted. Furthermore, our data are limited to nine urban cities of Pakistan which largely exclude data from rural villages. In addition, because our study was limited to non-medical women, the pooled prevalence of screening practices did not include all existing literature. Furthermore, the non-probability sampling methods used in some included studies might limit the generalisability of our results.

## 5. Conclusion

In conclusion, the level of BCa awareness is alarmingly low amongst Pakistani women. Approximately, only two in five women are aware of at least one BCa risk factor, symptom, or diagnostic modality. Less than two in five women appreciate mammography as a diagnostic tool. Despite a good appreciation of BCa and its screening methods, less than two in five women practice regular BSE, and less than one in five women have ever undergone a CBE. In light of our findings, the need for nationwide awareness programs is most pressing as Pakistan shoulders the highest BCa-associated global death rate.

## Supporting information

**S1 File. PRISMA 2020 checklist.** This file contains the Preferred Reporting Items for Systematic reviews and Meta-Analyses (PRISMA) checklist.
(PDF)

**S2 File. Search string(s).** This file contains the individual search strings used for each database to perform the literature search for this study.
(PDF)

**S3 File. Quality analysis.** This file contains a detailed quality analysis using the National Institute of Health's Quality Assessment Tool for Observational Cohort and Cross-Sectional Studies.
(PDF)

**S4 File. Further meta-analyses results.** This file contains further meta-analysis results, including funnel plots, and tests for assessment of publication bias.
(PDF)

## Author Contributions

**Conceptualization:** Muhammad Abdul Rehman, Eilaf Ahmed Meenai.

**Data curation:** Muhammad Abdul Rehman, Erfa Tahir, Huzaifa Ghulam Hussain, Ayesha Khalid, Syed Mohammad Taqi, Eilaf Ahmed Meenai.

**Formal analysis:** Muhammad Abdul Rehman.

**Methodology:** Muhammad Abdul Rehman, Erfa Tahir.

**Supervision:** Muhammad Abdul Rehman.

**Validation:** Muhammad Abdul Rehman.

**Visualization:** Muhammad Abdul Rehman, Huzaifa Ghulam Hussain.

**Writing – original draft:** Muhammad Abdul Rehman, Erfa Tahir, Huzaifa Ghulam Hussain, Ayesha Khalid, Syed Mohammad Taqi.

**Writing – review & editing:** Muhammad Abdul Rehman, Erfa Tahir.

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
