## [Decision Letter · Decision Letter 0]

27 Dec 2023

PONE-D-23-38477Awareness Regarding Breast Cancer Amongst Women in Pakistan: A Systematic Review and Meta-AnalysisPLOS ONE

Dear Dr. Abdul Rehman,

Thank you for submitting your manuscript to PLOS ONE. After careful consideration, we feel that it has merit but does not fully meet PLOS ONE’s publication criteria as it currently stands. Therefore, we invite you to submit a revised version of the manuscript that addresses the points raised during the review process. Please revise your manuscript in accordance with the reviewers' comments and suggestions to enhance its clarity, rigor, and overall quality.

We look forward to receiving your revised manuscript.

Kind regards,

Sina Azadnajafabad, MD, MPH

Academic Editor

PLOS ONE

Journal Requirements:

2. We note that Figure 2 in your submission contain [map/satellite] images which may be copyrighted. All PLOS content is published under the Creative Commons Attribution License (CC BY 4.0), which means that the manuscript, images, and Supporting Information files will be freely available online, and any third party is permitted to access, download, copy, distribute, and use these materials in any way, even commercially, with proper attribution. For these reasons, we cannot publish previously copyrighted maps or satellite images created using proprietary data, such as Google software (Google Maps, Street View, and Earth). For more information, see our copyright guidelines: http://journals.plos.org/plosone/s/licenses-and-copyright.

Reviewers' comments:

Reviewer's Responses to Questions

**Comments to the Author**

1. Is the manuscript technically sound, and do the data support the conclusions?

Reviewer #1: Partly

Reviewer #2: Yes

2. Has the statistical analysis been performed appropriately and rigorously? 

Reviewer #1: Yes

Reviewer #2: Yes

3. Have the authors made all data underlying the findings in their manuscript fully available?

Reviewer #1: Yes

Reviewer #2: Yes

4. Is the manuscript presented in an intelligible fashion and written in standard English?

Reviewer #1: Yes

Reviewer #2: Yes

5. Review Comments to the Author

Reviewer #1: Dear Authors

Thank you for submitting the article for the possible consideration for publication. The area of your research was interesting and may contribute to the available knowledge of breast cancer which is one of the prevalent and high risks for most of women in the world. However, the following concerns are raised to improve the article.

Comments by sections

1. Title

Awareness Regarding Breast Cancer amongst Women in Pakistan: A Systematic Review and Meta-Analysis.

-Better to use status/level of awareness?

2. Abstract

-Not use abbreviation on abstract part

-Is knowledge and awareness similar? Your title was awareness but the associated factors mentioned for awareness. Make them consistent

-It lacks recommendation

Introduction

• I think you have three objectives. But, nothing mentioned on your introduction parts regarding your objectives. For example, regarding risk factors, treatment, attitude, screening practices.

• Please re-write the introduction parts magnitude, burden, risk factors, possible interventions and major gaps.

Materials and methods

Inclusion and exclusion criteria

• What about knowledge of breast cancer. Were they excluding form the search?

• Why you exclude the medical students/ physician. You are searching for awareness of BCa. Please elaborate the reason.

• On your document you used knowledge and awareness interchangeably. Please make it consistent.

• You said observational studies. Which means your study include case-control, cohort and cross-sectional. Please specify observational.

Discussion

• Please see again your discussion parts .it only focus on magnitude rather than factors.

Reviewer #2: Thank you for your interesting and informing manuscript.

1-I will appreciate it if you could explain more about the method of weight assigning which is described in statistical analysis.

2- What are your suggestions to improve breast cancer knowledge in women in Pakistan?

6. PLOS authors have the option to publish the peer review history of their article (what does this mean?). If published, this will include your full peer review and any attached files.

Reviewer #1: No

Reviewer #2: No

---

## [Author Response · Author response to Decision Letter 0]

8 Jan 2024

PONE-D-23-38477 Awareness Regarding Breast Cancer Amongst Women in Pakistan: A Systematic Review and Meta-Analysis

Response to Reviewers’ Comments

5th of January, 2024

I would first like to thank the reviewers for their valuable comments, and their willingness to dedicate their time towards our work. 

Secondly, I would like to convey a description of the format of this document so as to make it easy, convenient, and smooth for the editor and the reviewers to understand how the comments have been addressed. Each reviewer’s comment is followed by a response. The answers highlight line numbers, and in quotations, the change that was made. I have also made use of evidence through links to support my responses where necessary. 

Responses to Comments by the Editor

[1] In regard to the geographical map in our study, I would like to inform the editorial team that the original figure is freely available under the Creative Commons Attribution-Share Alike 4.0 International license and is available here: https://en.m.wikipedia.org/wiki/File:Pakistan_provinces_2018_HDI_map.svg

I have updated the source of the original figure in the section Method>Illustrations. We feel that because this figure is publicly available, it does not require a signed consent. However, I may be mistaken given the technicalities of the license etc. If that is so, do let us know so we may make the appropriate changes. 

[2] The revised manuscript has been formatted according to Plos One's style. All associated files have been as well. 

Responses to Comments by Reviewer #1 

Comment 1

”1. Title: Awareness Regarding Breast Cancer amongst Women in Pakistan: A Systematic Review and Meta-Analysis. -Better to use status/level of awareness?”

Response 1

When drafting the title for our manuscript, we put forward several recommendations, including the ones suggested by the reviewer. Upon subsequent revisions, and proofreading, we shrunk the title to keep it concise, to the point, and pertinent. While “Status of awareness…” or “Level of awareness…”, are also correct when it comes to grammar and written English, restricting the title to “Awareness…” avoids the superfluous use of words. This was our rationale for not choosing “Level of awareness…” in the title. 

Furthermore, there are several published articles in high-impact factor journals that make use of similar titles, which highlights the scientific, and grammar-related accuracy of the title. Some examples are as follows: 

A) https://pubmed.ncbi.nlm.nih.gov/34227943/ - Awareness and prevalence of needle stick injuries among cleaners and health-care providers in Gaza Strip hospitals: a cross-sectional study

B)https://pubmed.ncbi.nlm.nih.gov/10703802/ - Awareness during anaesthesia: a prospective case study

Comment 2

2.Abstract -Not use abbreviation on abstract part

Response 2

Firstly, I would like to respond to this comment by highlighting the limited word count that is universally kept for all abstracts. Therefore, it is not feasible to not use acronyms, as it eats away at the total word count which can be used to convey the scientific information that an abstract needs to convey. I presume this is the reason why Plos One allows the use of abbreviations where they can not be avoided (Plos One guide: https://journals.plos.org/plosone/s/submission-guidelines#loc-manuscript-organization). 

Secondly, all the abbreviations used in the abstract have already been defined at their first use.

Thirdly, although abbreviations that are universally known, often do not require an expanded mention in the title or the abstract, which is in consideration of the limited word count. Yet, we have chosen to expand the following abbreviations: “PRISMA” is now “Preferred Reporting Items for Systematic Reviews and Meta-Analyses” (Line 18-19); “NIH” is now “National Institutes of Health” (Line 23). Some examples of abstracts in published literature where abbreviations have been used and not defined at first use are: 

A)https://pubmed.ncbi.nlm.nih.gov/35247352/ - Daily steps and all-cause mortality: a meta-analysis of 15 international cohorts [CI is not defined at all]

B)https://pubmed.ncbi.nlm.nih.gov/33635310/ - Association of Convalescent Plasma Treatment With Clinical Outcomes in Patients With COVID-19: A Systematic Review and Meta-analysis [COVID, CI not defined at all]

Comment 3

-Is knowledge and awareness similar? Your title was awareness but the associated factors mentioned for awareness. Make them consistent

Response 3

Because the objective of our study was to evaluate the prevalence of knowledge and practice (Lines 16-17, Lines 68-71), we chose an overarching, umbrella synonym that covers both knowledge and practice. Similarly, our abstract goes on to report results which are consistent with these objectives (Lines 25-31).

I appreciate this comment, and would like to point out that to make the distinction between “awareness” and “knowledge”, and to highlight the overarching nature of the word “awareness”, the manuscript already contains an explanatory sentence in the Method section (Lines 105-106: “The following data regarding BCa awareness were sought: 1) knowledge about BCa, 2) attitude towards BCa, 3) practice towards BCa.”)

Lastly, we have changed “…low levels of BCa awareness: risk factors, 42.7%…” to “…low levels of BCa knowledge: risk factors, 42.7%…” (Line 26).

Comment 4

-It lacks recommendation

Response 4

I would like to point out that the PRISMA 2020 for Abstracts checklist (which can be found here: http://www.prisma-statement.org/extensions/Abstracts) does not list adding recommendations as one of its components. Similarly, the Plos One guide (which can be found here: https://journals.plos.org/plosone/s/submission-guidelines#loc-manuscript-organization) also does not list adding recommendations to the abstract. Furthermore, the abstract is limited to 300 words, which does not allow us to stray from the guidelines. 

Comment 5

Introduction I think you have three objectives. But, nothing mentioned on your introduction parts regarding your objectives. For example, regarding risk factors, treatment, attitude, screening practices.

Response 5

Our research indeed has three objectives, which is to consolidate data about knowledge, attitude, and practice regarding breast cancer in Pakistan (in short: awareness regarding breast cancer; as is conveyed by the title and the explanation in the Method section). 

I would like to point out the study design of our work which is a systematic review/meta-analysis, and not a primary study. Our aim is to synthesize evidence which could guide future health policies, and highlight the current situation in Pakistan. Our aim is not to compare one intervention/treatment/risk factor with another; which, if it were, would require that we describe and introduce each intervention. Therefore, we focused our introduction on the overall awareness of breast cancer. 

Furthermore, while we could certainly quote data regarding awareness in the introduction, we decided that it would be best to keep such information within the confines of the Results section, which satisfies the purpose of our systematic review. 

In order to satisfy the suggestion, we have added another paragraph to highlight why our research is a timely need where we highlight the importance of knowledge about BCa-related interventions/risk factors etc. (Lines 63-70)

Comment 6

• Please re-write the introduction parts magnitude, burden, risk factors, possible interventions and major gaps.

Response 6

Unfortunately, I fail to grasp why these sections need to be rewritten. Perhaps the reviewer meant that they be added. In order to address this comment I will point out the line numbers of each section:

Magnitude and burden: Lines 37-47

Risk Factors: Line 68

Interventions: Lines 65-67

Major Gap: Lines 72-73

Comment 7

Materials and methods. Inclusion and exclusion criteria. • What about knowledge of breast cancer. Were they excluding form the search?

Response 7

No, such studies were not excluded. In fact, assessing knowledge is a core objective of our study. The term “awareness” (Line 93) includes studies that assessed knowledge. 

Comment 8

• Why you exclude the medical students/ physician. You are searching for awareness of BCa. Please elaborate the reason.

Response 8

We excluded medical students and physicians because assessing their knowledge and including it in the statistical analysis would significantly skew the results by introducing a massive knowledge bias. This is so because medical students and physicians are taught about breast cancer thoroughly and meticulously which makes an assessment of their knowledge almost irrelevant when gauging the current awareness in Pakistan.

Comment 9

• On your document you used knowledge and awareness interchangeably. Please make it consistent.

Response 9

As explained previously and in the Method section (Lines 112-113), “awareness” is an umbrella term which is includes knowledge, attitude etc. In our manuscript, we have used them accordingly. The term “knowledge” is used wherever knowledge is calculated or referred to. “Awareness” is used wherever the overall results of our work are referred to.

Comment 10

• You said observational studies. Which means your study include case-control, cohort and cross-sectional. Please specify observational.

Response 10

We have already specified the type of observational studies in the Methods section (Line 92): “We included observational cross-sectional studies which were…”

Comment 11

Discussion • Please see again your discussion parts .it only focus on magnitude rather than factors.

Response 11

Firstly, I would like to point out the same response as was given for Comment 5: our aim is to consolidate data and not comment on individual interventions/risk factors/treatments etc. 

Secondly, the discussion does comment on various pertinent and relevant factors such as self-examination which is a primary screening method that is hailed as cheap and effective (Lines 285-293). Similarly, mammography (Lines 278-283), discussion of factors for poor knowledge (Lines 304-314), comparison with neighboring countries and literature (Lines 316-323), and a discussion of the statistical interpretation (Lines 325-345) have been discussed in detail. This is in addition to the prevalence/magnitude which open the discussion in the introductory paragraph. 

Responses to Comments by Reviewer #2 

Comment 12

1-I will appreciate it if you could explain more about the method of weight assigning which is described in statistical analysis.

Response 12

We assigned weights under a random effects model (Line 132) to account for the varying heterogeneity in the data across studies owing to various factors such as geography, socioeconomic status etc., as is conventional for single proportion meta-analyses which pool data from observational studies. In particular, we used the DerSimonian random effects model which was introduced in 1986, accounts for inter-study variance, and has been amongst one of the most widely used models for large data in single proportion meta-analyses. It makes use of inverse weights, which is utilized in meta-analyses calculating prevalence, or similar.

Links for further reading and the use of the model in other single proportion meta-analyses can be found here: 

A)https://pubmed.ncbi.nlm.nih.gov/3802833/ - Meta-analysis in clinical trials

B)https://pubmed.ncbi.nlm.nih.gov/16807131/ - Random-effects model for meta-analysis of clinical trials: an update

C)https://www.ncbi.nlm.nih.gov/pmc/articles/PMC2667312/ - A re-evaluation of random-effects meta-analysis

D)https://pubmed.ncbi.nlm.nih.gov/36000793/ - Prevalence and risk factors of self-reported psychotic experiences among high school and college students: A systematic review, meta-analysis, and meta-regression

E)https://pubmed.ncbi.nlm.nih.gov/33958198/ - A meta-analysis and meta-regression on the prevalence of lipohypertrophy in diabetic patients on insulin therapy

F)https://www.ncbi.nlm.nih.gov/pmc/articles/pmid/37131226/ - Malaria prevalence in Mauritania: a systematic review and meta-analysis

Comment 13

2- What are your suggestions to improve breast cancer knowledge in women in Pakistan?

Response 13

Our recommendations/suggestions to improve breast cancer-related knowledge is primarily through organized awareness campaigns, for which our work lays the stepping stone. We have described their effectiveness in the introduction (Lines 62-70), discussion (Lines 311-315), and conclusion (Lines 361-363).

I hope our responses are satisfactory for the reviewers. Incase further explanations are needed or suggestions are available, I would gladly receive them. I want to once again thank both reviewers for their comments.

Warm Regards, 

Dr. Abdul Rehman

---

## [Decision Letter · Decision Letter 1]

14 Jan 2024

PONE-D-23-38477R1Awareness Regarding Breast Cancer Amongst Women in Pakistan: A Systematic Review and Meta-AnalysisPLOS ONE

Dear Dr. Rehman,

Thank you for submitting your manuscript to PLOS ONE. After careful consideration, we feel that it has merit but does not fully meet PLOS ONE’s publication criteria as it currently stands. Therefore, we invite you to submit a revised version of the manuscript that addresses the points raised during the review process.

We look forward to receiving your revised manuscript.

Kind regards,

Sina Azadnajafabad, MD, MPH

Academic Editor

PLOS ONE

Journal Requirements:

Reviewers' comments:

Reviewer's Responses to Questions

**Comments to the Author**

1. If the authors have adequately addressed your comments raised in a previous round of review and you feel that this manuscript is now acceptable for publication, you may indicate that here to bypass the “Comments to the Author” section, enter your conflict of interest statement in the “Confidential to Editor” section, and submit your "Accept" recommendation.

Reviewer #1: All comments have been addressed

Reviewer #2: All comments have been addressed

2. Is the manuscript technically sound, and do the data support the conclusions?

Reviewer #1: Yes

Reviewer #2: Yes

3. Has the statistical analysis been performed appropriately and rigorously? 

Reviewer #1: Yes

Reviewer #2: Yes

4. Have the authors made all data underlying the findings in their manuscript fully available?

Reviewer #1: Yes

Reviewer #2: Yes

5. Is the manuscript presented in an intelligible fashion and written in standard English?

Reviewer #1: Yes

Reviewer #2: Yes

6. Review Comments to the Author

Reviewer #1: Thank you for submitting the article for the possible consideration for publication. The area of your research was interesting and may contribute to the available knowledge of breast cancer which is one of the prevalent and high risks for most of women in the world. All comments were addressed. But, still your manuscript needs some grammar correction issue.

• You mentioned on response cross sectional study. But, throughout the manuscript I didn’t seen cross sectional study. So, please specify observational.

Reviewer #2: (No Response)

7. PLOS authors have the option to publish the peer review history of their article (what does this mean?). If published, this will include your full peer review and any attached files.

Reviewer #1: No

Reviewer #2: **Yes: **Ghazal Daftari

---

## [Author Response · Author response to Decision Letter 1]

14 Jan 2024

PONE-D-23-38477 

Awareness Regarding Breast Cancer Amongst Women in Pakistan: A Systematic Review and Meta-Analysis

Response to Reviewers’ Comments

Round Two

15th of January, 2024

Once again, I would like to begin by first thanking the reviewers who dedicated time out of their busy schedules towards our work. 

Secondly, I would like to convey a description of the format of this document so as to make it easy, convenient, and smooth for the editor and the reviewers to understand how the comments have been addressed. Each reviewer’s comment is followed by a response. The answers highlight line numbers, and in quotations, the change that was made. I have also made use of evidence through links to support my responses where necessary. 

Responses to Comments by Reviewer #1 

Comment 1 

Thank you for submitting the article for the possible consideration for publication. The area of your research was interesting and may contribute to the available knowledge of breast cancer which is one of the prevalent and high risks for most of women in the world. All comments were addressed. But, still your manuscript needs some grammar correction issue.

• You mentioned on response cross sectional study. But, throughout the manuscript I didn’t seen cross sectional study. So, please specify observational.

Response 1

I believe that perhaps some oversight has been made, as this comment was addressed in the first round of revisions. Nonetheless, I will point out the locations where we already specify the type of observational study design that was included In our research in order to provide maximum aid and convenience.

We specify the type of observational study design on the following line numbers:

1.Line 91, Methods>Inclusion/exclusion criteria: “We included observational cross-sectional studies that were published in/after 2010…” 

2.Line 151. Results>Characteristics of Included Studies: “All included studies were observational cross-sectional studies.”

3.Line 163, Table 1, where each study is specified as having a cross-sectional study design. 

Responses to Comments by Reviewer #2 

No comments have been raised by Reviewer #2 in this round.

I hope my responses is satisfactory for Reviewer 1. Incase further explanations are needed, or suggestions are available, I would gladly receive them. I want to once again thank both reviewers for their comments.

Warm Regards, 

Dr. Abdul Rehman

---

## [Editor Report · Decision Letter 2]

23 Jan 2024

Awareness Regarding Breast Cancer Amongst Women in Pakistan: A Systematic Review and Meta-Analysis

PONE-D-23-38477R2

Dear Dr. Rehman,

We’re pleased to inform you that your manuscript has been judged scientifically suitable for publication and will be formally accepted for publication once it meets all outstanding technical requirements.

Kind regards,

Sina Azadnajafabad, MD, MPH

Academic Editor

PLOS ONE
---

## [Editor Report · Acceptance letter]

27 Feb 2024

PONE-D-23-38477R2 

PLOS ONE

Dear Dr. Abdul Rehman, 

I'm pleased to inform you that your manuscript has been deemed suitable for publication in PLOS ONE. Congratulations! Your manuscript is now being handed over to our production team.

Kind regards, 

on behalf of

Dr. Sina Azadnajafabad 

Academic Editor

PLOS ONE